# Plasma hsa-miR-22-3p Might Serve as an Early Predictor of Ventricular Function Recovery after ST-Elevation Acute Myocardial Infarction

**DOI:** 10.3390/biomedicines11082289

**Published:** 2023-08-17

**Authors:** Liana Maries, Alexandra Ioana Moatar, Aimee Rodica Chis, Catalin Marian, Constantin Tudor Luca, Ioan-Ovidiu Sirbu, Dan Gaiță

**Affiliations:** 1Biochemistry Department, “Victor Babes” University of Medicine and Pharmacy, 300041 Timisoara, Romania; maries.liana@umft.ro (L.M.); moatar.alexandra@umft.ro (A.I.M.); chis.aimee@umft.ro (A.R.C.); cmarian@umft.ro (C.M.); 2Doctoral School, “Victor Babes” University of Medicine and Pharmacy, 300041 Timisoara, Romania; 3Center for Complex Network Science, “Victor Babes” University of Medicine and Pharmacy, 300041 Timisoara, Romania; 4Cardiology Department, “Victor Babes” University of Medicine and Pharmacy, 300041 Timisoara, Romania; costiluca67@yahoo.ro (C.T.L.); dgaita@cardiologie.ro (D.G.); 5Research Center of the Institute of Cardiovascular Diseases Timisoara, 13A Gheorghe Adam Street, 300310 Timisoara, Romania

**Keywords:** acute myocardial infarction, left ventricular remodeling, microRNA, diabetes mellitus, miR-22

## Abstract

Left ventricle remodeling (LVR) after acute myocardial infarction (aMI) leads to impairment of both systolic and diastolic function, a major contributor to heart failure (HF). Despite extensive research, predicting post-aMI LVR and HF is still a challenge. Several circulant microRNAs have been proposed as LVR predictors; however, their clinical value is controversial. Here, we used real-time quantitative polymerase chain reaction (qRT-PCR) to quantify hsa-miR-22-3p (miR-22) plasma levels on the first day of hospital admission of ST-elevation aMI (STEMI) patients. We analyzed miR-22 correlation to the patients’ clinical and paraclinical variables and evaluated its ability to discriminate between post-aMI LVR and non-LVR. We show that miR-22 is an excellent aMI discriminator and can distinguish between LVR and non-LVR patients. The discriminative performance of miR-22 significantly improves the predictive power of a multiple logistic regression model based on four continuous variables (baseline ejection fraction and end-diastolic volume, CK-MB, and troponin). Furthermore, we found that diabetes mellitus, hematocrit level, and the number of erythrocytes significantly influence its levels. These data suggest that miR-22 might be used as a predictor of ventricular function recovery in STEMI patients.

## 1. Introduction

Although the incidence of acute myocardial infarction (aMI) with ST-elevation (STEMI) has significantly decreased in recent decades, aMI is still one of the leading causes of morbidity and mortality [1,2]. The clinical evolution post-aMI is influenced by multiple factors, but the most feared complications are left ventricular remodeling (LVR) and alteration of left ventricular function (LVF) [3,4]. A decrease as low as 5% in left ventricular ejection fraction (LVEF) is associated with a significant increase in post-aMI mortality, and improvement of LVF after aMI is associated with an improved prognosis and reduced mortality [5,6,7].

Several first-line predictors of LVF recovery have been advanced, of which LVEF evolution, cardio-respiratory arrest or ventricular fibrillation, maximal troponin level, and high leukocytosis performed the best [8,9,10]. Second-line predictors such as infarct size, myocardial salvage index, and myocardial viability have been introduced with the advent of new generation imaging techniques such as cardiac magnetic resonance (CMR), single photon emission computed tomography (SPECT), positron emission tomography or dobutamine stress echocardiography [11,12]. Presumably, a combination of circulant biomarkers for myocardial necrosis and inflammatory response rather than individual biomarkers might be used as LVR and LVF recovery predictors [13].

The individual prognostic value of these predictors is rather modest and still needs to be explored and understood; their performance is influenced by demographics, comorbidities, associated therapies, moment of sampling, observer bias, imaging quality, and techniques [14]. Therefore, the need for methods and biomarkers able to reliably predict LVF dysfunction and LVR is still of utmost importance [15].

MicroRNAs (miRNAs) are small RNA molecules that post-transcriptionally regulate the expression of target genes, being involved in all aspects of cardiac pathology, including aMI and LVR. The time-dependent regulation of microRNAs is pivotal for a favorable post-aMI clinical response [16,17].

While multiple circulant microRNAs have been advanced as post-aMI LVR predictors, the data on miRNAs association with ventricular recovery after aMI are scarce [18]. MiR-499a and miR-125b have been proposed as predictors (both AUCs around 0.74) of LVF recovery in heart failure drug-refractory patients after cardiac resynchronization therapy [19]. Plasma level of miR-1 at admission positively correlates with variations of end-diastolic volume (EDV) and end-systolic volume (ESV); however, its prognostic value was rather modest, with an AUC < 0.7 [20]. The combined discharge levels of four circulant microRNAs (miR-16/27a/101/150) were found to improve the prediction of LV contractility post-aMI [21]. Day of admission circulant miR-1 and miR-133a correlate positively with high-sensitivity cardiac troponin T (hsTnT) and negatively with LVEF, while the combination of N-terminal prohormone of brain natriuretic peptide (NT-proBNP) and miR-499 identified an “at risk” group for LVR [22]. In a small group of percutaneous coronary intervention (PCI)-treated aMI patients, plasma miR-143 was found to modestly correlate with the change of LVEF at six months post-aMI [23].

miR-22-3p (miR-22) is one of the most abundantly expressed microRNAs in the heart. Circulant miR-22 is significantly upregulated in the acute phase of aMI and correlates to (and peaks earlier than) cardiac troponin I (cTnI) [24,25]. Expression of heart miR-22 is altered in ischemia/reperfusion models, strongly depending on reperfusion [26,27]. However, data on miR-22 association with aMI are conflicting since it decreased in blood samples collected before PCI [28]. Furthermore, it is unclear which of the 3p and 5p forms of miR-22 play a role in the ischemia/reperfusion myocardial events and contribute to aMI physiopathology and clinical evolution.

The present study uses real-time quantitative PCR to investigate plasma hsa-miR-22-3p in the clinical context of aMI, characterize its association with the clinical and paraclinical variables, and investigate its potential as a predictor of ventricular recovery post-STEMI.

## 2. Materials and Methods

### 2.1. Inclusion and Exclusion Criteria

The study was conducted in accordance with the Declaration of the Helsinki Code of Ethics and has been reviewed and approved by the ethics review board (3822/31 May 2016). The study took place at the Institute for Cardiovascular Disease in Timisoara, Romania, and involved day-of-admission blood sampling and echocardiographic evaluations. We enrolled 105 patients diagnosed with STEMI; all the patients provided signed, written consent.

The STEMI diagnostic followed the guidelines issued by the European Society of Cardiology (ESC). We included only adult patients admitted to the hospital within 12 h from aMI symptoms onset and for whom pre-hospital care was conducted according to the ESC guidelines for aMI with ST-elevation (antiplatelet: Aspirin 300 mg, Ticagrelor 180 mg or Clopidogrel 600 mg, Atorvastatin 80 mg, and fibrinolytic therapy and anticoagulation in selected patients).

Patients with a history of any form of coronary artery disease, resuscitated cardiac arrest prior to hospital admission, associated diagnostics of cancer, acute infectious diseases, and liver dysfunction, or unable to provide a signed written consent were excluded.

All patients underwent primary, routine early, or rescue PCI, as per ESC guidelines.

The control group included 17 outpatients without any medical history of aMI, for which an aMI diagnostic was excluded, and all the exclusion criteria apply.

### 2.2. The Follow-Up Group

Only 43 patients participated in the follow-up echocardiographic evaluation one year post-aMI. LVR was diagnosed if the patients developed LV dysfunction and heart failure (HF) symptoms, with a minimum of 10% increase from the baseline in end-diastolic volume (ΔEDV) and LVEF < 50% (*n* = 14). In the non-LVR group, the patients did not develop LV dysfunction and HF symptoms, LVEF  ≥  50%; ΔEDV < 10% (*n*  =  29). Killip classification of STEMI patients and the burden of coronary artery disease (CAD) was calculated according to the guidelines issued by the ESC.

### 2.3. Specimen Collection

From each patient, we collected 3 mL of blood in EDTA-coated tubes from the antecubital vein. Plasma was separated (10 min centrifugation at 1500 rpm and room temperature) within ten minutes and stored at −80 °C until further use. All blood samples were collected on admission day, before PCI. We discarded all samples showing signs of hyperlipemia, hemolysis (verified spectrophotometrically by measuring absorbance at 414 nm), and icterus.

### 2.4. RNA Purification

We used the miRNeasy Serum/Plasma Kit (Qiagen, Hilden, Germany, catalog no. 217184) to isolate total RNA from 200 µL of plasma and used a Nanodrop 2000 instrument to check the RNA quality and quantity (A260/A280 and A230/A260 ratios).

### 2.5. PCR Detection

All RNA samples were spiked in with *Caenorhabditis elegans* cel-miR-39 as an external normalizer. cDNA was synthesized starting from equal (10 ng) total RNA inputs using the TaqMan^®^ MicroRNA Reverse Transcription Kit (Applied Biosystems, Waltham, MA, USA, catalog no. 4366596). All qRT-PCR reactions were performed in duplicate using inventoried TaqMan™ MicroRNA Assays (ThermoFisher Scientific, Waltham, MA, USA, assays ID 000398 and 000200). Fold changes were calculated using the ΔΔCT method of relative quantification with cel-miR-39 as a normalizer [29].

### 2.6. Statistical Analysis

We used Prism 9 for MacOS, Version 9.3.1, to perform all statistical analyses. Basic descriptive statistics were used for demographic, clinical, functional, and laboratory data of the patients; we used the Kolmogorov–Smirnov test to check data distribution. The differences between continuous variables were assessed using the heteroscedastic Student’s *t*-test (for variables with normal distribution) and the Mann–Whitney U test (for data not normally distributed). We used the Z-test to compare the binary variables datasets. Correlation analyses for continuous data sets were performed using either the Spearman test or the Pearson test (for normally distributed data sets). Correlation between binary variables and continuous variables was tested using the point biserial test. For all tests, the threshold of statistical significance is 0.05. All statistical tests are two-tailed.

The flowchart depicting the rationale of the study is presented in Figure 1.

## 3. Results

### 3.1. Baseline Clinical Data of Patients

The demographic features and clinical data for the patients included in our analysis are shown in Table 1. The median age of the cohort was 61 (minimal 29, maximal 87) years, and 27.62% of the patients were females. The most prevalent risk factors were arterial hypertension (71.4%) and smoking (50.48%). The most frequent MI locations were anterior (46.7%) and inferior (48.6%), and the median interval between symptom onset to reperfusion was 6 (1.5–16) hours; 7 patients (6.7%) died. All STEMI and control patients’ data can be accessed in Appendix A.

Table 2 summarizes the demographic, clinical, and echocardiographic parameters of the two follow-up subgroups stratified based on the presence of LV remodeling. Except for the changes in ejection fraction and end-diastolic volumes, there are no significant differences between the LVR and non-LVR subgroups. All echocardiographic data of the follow-up group can be accessed in Appendix A.

### 3.2. miR in STEMI Patients vs. Control

Compared to the control group, miR-22 plasma levels are significantly increased in STEMI patients compared to the control group (*p* = 0.001) (Figure 2A). The receiver operating characteristic (ROC) curve analysis showed that miR-22 is an excellent early discriminator of STEMI from controls, confirming that it could be used as a diagnostic biomarker for aMI (Figure 2B).

### 3.3. miR and Clinical Parameters in STEMI Patients

We systematically assessed the correlation of miR-22 adjusted Ct values to the clinical and paraclinical parameters accessible through the electronic hospital archive. There is a statistically significant correlation to diabetes mellitus (DM), erythrocyte number, and hematocrit (Table 3).

### 3.4. miR in the Follow-Up Group

There is no statistically significant difference between miR-22 plasma values in the STEMI and the follow-up cohorts (*p* = 0.9852, Welch’s *t*-test); however, follow-up plasma miR-22 values are no longer correlated to DM, the erythrocyte number, and the hematocrit (Table 3). There is a statistically significant difference between the plasma miR-22 levels in the LVR vs. non-LVR *p* (FC = 2.37, *p* = 0.0322, Welch’s *t*-test) (Figure 3A). The ROC curve analysis shows that early miR-22 is a good discriminator of LVR from non-LVR patients, indicating that it could be used as an LVR predictive biomarker for STEMI patients (AUC = 0.7266, 95% CI 0.5632 to 0.89, *p* = 0.0171). Of note, plasma miR-22 negatively correlates to %ΔEF in the non-LVR but not in the LVR group; there is no correlation to %ΔEDV in either group (Table 4).

A multiple logistic regression model built using LVR as outcome categorical variable and miR-22, initial EF (EFi), initial EDV (EDVi), CK-MB, and troponin as continuous variables shows that the inclusion of miR-22 into the model (odds ratio 3.036; 95% confidence interval: 1.437 to 10.68, *p* = 0.0203) significantly improved both the sensitivity and specificity of discrimination (Table 5, Figure 3B).

### 3.5. miR in Diabetic vs. Non-Diabetic Patients

Given the correlation to DM (known for its impact on post-aMI LV dynamics), we investigated whether DM could influence miR-22 performance as an aMI and LVR discriminator. There are no significant differences in terms of demographics, cardiovascular risk factors, and type of myocardial infarction between the diabetic and non-diabetic patients included in the present study (Table 6). However, there is a significant discrepancy in hypertension (*p* = 0.048) and in-hospital death ratios (*p* = 0.0015) in favor of diabetic patients.

In diabetic STEMI patients, plasma miR-22 is downregulated compared to the non-diabetic STEMI group (fold change FC = 0.51; *p* = 0.0165); however, both diabetic and non-diabetic STEMI plasma levels of miR-22 are strongly upregulated compared to controls (FC = 17.6, *p* = 0.0006 and FC = 34.5, *p* < 0.001, respectively) (Figure 4A). Of note, the diabetic status significantly influences the ability of plasma miR-22 to differentiate between aMI and controls (Figure 4B). 

## 4. Discussion

Given their outstanding stability, circulant microRNAs have drawn attention as possible diagnostic and predictive biomarkers in various pathologies, including aMI. However, the results from aMI research are non-overlapping, at times contradictory, mainly due to differences in study design, methodologies, the definition of post-aMI adverse events, the timing of sampling, and the analytical tools used.

Our data on early (hospital admission day) plasma hsa-miR-22-3p confirm its differential expression in STEMI patients vs. controls and support its use as a putative diagnostic aMI STEMI biomarker. Of note, the aMI discriminative performance of early plasma miR-22 is marginally affected by DM.

We also show that plasma miR-22-3p is differentially expressed and efficiently discriminates between LVR and non-LVR patients. Moreover, plasma miR-22 significantly improves the LVR discriminative performance of a multiple logistic regression model based on four continuous variables (EFi, EDVi, CK-MB, and troponin). Surprisingly, plasma miR-22 levels only correlate to EF changes in the non-LVR patients; the changes in EDV are not correlated to miR-22 in either group. This could indicate that miR-22 might predict ventricular function recovery rather than LVR.

The evolution of both young and older STEMI patients is complicated by ventricular systolic dysfunction [4,30]. Post-aMI recovery of ventricular function positively correlates to higher baseline EF and lower peak troponin (both indicators of MI size) [8,31]. Of note, STEMI patients are more likely to show abnormal EF and higher troponin levels; the recovery of their ventricular function occurs more often in patients with fewer comorbidities and significantly reduces all-cause and cardiovascular mortality [30].

We have assessed the ventricular function early in the evolution of STEMI; thus, both the myocardial stunning (reversible) and early necrosis (irreversible) of the ventricular wall might contribute to the echocardiographic evaluation results. Since there are no differences between the LVR and non-LVR cohorts regarding the therapy prescribed (ACE-I/ARB, beta-blockers, and MRA, to influence positively post-aMI recovery), one can assume that post-aMI therapy did not significantly influence the ventricular function outcome.

## 4.1. miR-22 and the Stressed Heart

Hsa-miR-22 is conserved in evolution, being transcribed from a unique host gene miR22HG located on 17p13.3; its expression is quasi-ubiquitous, enriched in the cardiomyocytes skeletal muscle cells and liver, being upregulated during myocyte differentiation [32,33,34]. In aged mice (and to a lesser extent in young mice), miR-22 inhibition prevented cardiac remodeling and significantly improved cardiac recovery after aMI [34]. In human patients over 60, circulant miR-22 is associated with cardiovascular mortality during a three-year follow-up study of ischemic and non-ischemic HF [34].

Both human and animal (ex vivo and in vivo) model data indicate that miR-22 is an essential component of the cardiac response to stress, modulating phenomena such as hypertrophy, fibrosis, oxidative stress, autophagy, and apoptosis [35].

Rat cardiomyocytes exposed (ex vivo and in vivo) to hypertrophic agonists respond by activating miR-22-dependent mechanisms that lead to the de-repression of fetal cardiac genes such as BNP and skeletal muscle α-actin [33]. Cultured rat cardiomyocytes exposed to phenylephrine or angiotensin II respond by upregulating miR-22 in correlation to hypertrophic markers; of note, the anti-hypertrophic effect of atorvastatin is mediated by a significant downregulation of miR-22 in both ex vivo and in vivo experiments on rodent cardiomyocytes [36,37].

Some miR-22-null mice develop cardiac developmental defects and die in utero while surviving adult miR-22 knock-out (KO) mice are prone to ventricular dilation upon long-term pressure overload [38]. However, cardiomyocyte-specific conditional miR-22 KO mice are viable and fertile, but their hearts have a reduced ability to adapt to stress manifested by exacerbated apoptosis and fibrosis, a progressive dilated cardiomyopathy, and a decrease in fractional shortening upon isoproterenol challenging [33]. This indicates that miR-22 is instrumental in preserving the ventricular function in stressed hearts. Of note, analysis of miR-22 KO mice suggests that the precise temporal balance of the microRNA response is important for cardiac post-stress recovery. The miR-22 upregulation seems to be an early response of the heart to pressure overload since in mice with transverse aortic constriction, the initial (first week) upregulation is followed by a steady decline to normal levels within the following four weeks [38]. Cardiac tissue-targeted overexpression of miR-22 in mice triggers the activation of a hypertrophic gene expression program that translates into significant cardiac dysfunction and heart failure [39].

However, the field is not free of controversies. In a rat model of short-term (2 h) post-aMI ischemia/reperfusion (IR), the early elevated intracardiac miR-22 aggravated the IR-induced cardiomyocyte mitochondrial damage, while in a 12 h post-aMI IR model, the intracardiac miR-22 was found decreased, and its overexpression significantly reduced the infarct size [26,27]. This timeline of miR-22 expression correlates with the data of Gurha et al. and points towards a time-dependent regulatory mechanism controlling miR-22 expression in the re-perfused myocardial tissue [38]. It also needs to be clarified to which extent the miR-22 effect involves cardiomyocyte apoptosis since the miR-22 overexpression results are conflicting [40].

An alternative explanation of the source of miR-22 post-aMI pertains to the well-known involvement of miR-22 in thrombus biology during the ischemia/reperfusion response. MiR-22 was found depleted in the thrombi extracted from aMI patients, suggesting it is released from activated platelets and uptaken by neighboring endothelia, where it targets Intercellular Adhesion Molecule 1 (ICAM-1) and facilitates the transfer of leukocytes into tissues [41]. This is significant since ischemic cardiomyocytes could uptake exosomes enriched in miR-22 in an ischemic preconditioning model of IR [42]. It is, thus, conceivable that the early upregulation of plasma miR-22 is the combined result of both platelet extrusion during thrombi formation and cardiomyocyte ischemic necrosis. Very recently, it has been shown that targeted depletion of platelet microRNAs in a mouse model of cardiac ischemia increased fibrosis, altered cardiac remodeling, and impaired LVF recovery [43]. How exactly the two sources (ischemic cardiomyocytes and the thrombi) contribute to the early elevation of plasma miR-22 and what the impact of miR-22 release on post-aMI physiopathology and LVR is remains to be established.

### 4.2. MicroRNAs and DM

Our STEMI group contains 24 people with diabetes, and their miR-22 plasma level is significantly decreased compared to non-diabetic STEMI patients. Although we found no correlation between plasma miR-22 and glycemia, the diabetic status per se impacts the ability of miR-22 to discriminate between aMI patients and controls. In patients with HF and aMI, DM was shown to influence LVR and LV dynamics, with a consecutive impact on their cardiovascular outcome [44,45]. Of note, in these patients, HbA1c values correlate negatively with the LVEF and strongly predict both short- and long-term complications [46,47,48].

Interestingly, miR-22-3p has recently emerged as a novel metabolic node, its targets being involved in lipid and glucose metabolism, insulin sensitivity, and thermogenesis. miR-22 regulates glucose homeostasis through Tcf7-mediated regulation of gluconeogenesis enzymes [49]. In vivo miR-22-3p antagonism improves insulin sensitivity, reduces circulant cholesterol and steatosis, and increases the level of non-esterified fatty acids [49,50].

### 4.3. Limitations

First, this is a single-center study with a relatively low number of participants; however, one should point out that we have designed (due to resource limitations) our approach as a small-scale pilot study, which awaits confirmation in a larger, multicenter trial. Second, less than 50% of the enrolled patients reported for evaluation at 12 months post-aMI. This might reflect a poor compliance of our patients that impedes the assessment of lifestyle and medication (like statins, antiplatelet drugs, beta blockers) impact on LVR [51,52]. Furthermore, we have limited our evaluation to 12 months post-aMI; thus, any further evolutions have yet to be noticed.

## 5. Conclusions

Post-aMI LVR associates sequentially coordinated changes in ventricular function, size, and shape, the consequence of apoptosis, hypertrophy, and fibrosis affecting the ventricular wall. The mechanical triggers (afterload and preload wall stress) are translated into a complex interplay of biochemical and molecular cues, including altered cardiomyocyte metabolism, activation of renin-angiotensin-aldosterone and sympathetic nervous systems, endothelin and extracellular matrix changes [53,54]. Since miR-22 upregulation (whether of cardiac or platelet origin) seems to be a rather early event, it is conceivable that it modulates the events related to the early stage of remodeling, such as apoptosis, cardiomyocyte hypertrophy, and extracellular matrix remodeling.

Many factors could impact the data on plasma microRNAs as aMI discriminators and LVR predictors, starting with the cohorts’ sample size, time of blood collection, microRNA detection and data normalization method, and LVR definition. Our data indicate that early (day of admission) plasma miR-22 might be used as aMI diagnostic and LVR predictor biomarker. miR-22 diagnostic performance is altered by DM status and correlates to paraclinical parameters such as the hematocrit and the erythrocytes number. While our study is exploratory in nature, it sets the stage for larger-scale investigations on plasma miR-22, expanding upon our preliminary results, that could ultimately pave the way for improved patient management and personalized interventions targeting ventricular remodeling post-aMI.

## Figures and Tables

**Figure 1 biomedicines-11-02289-f001:**
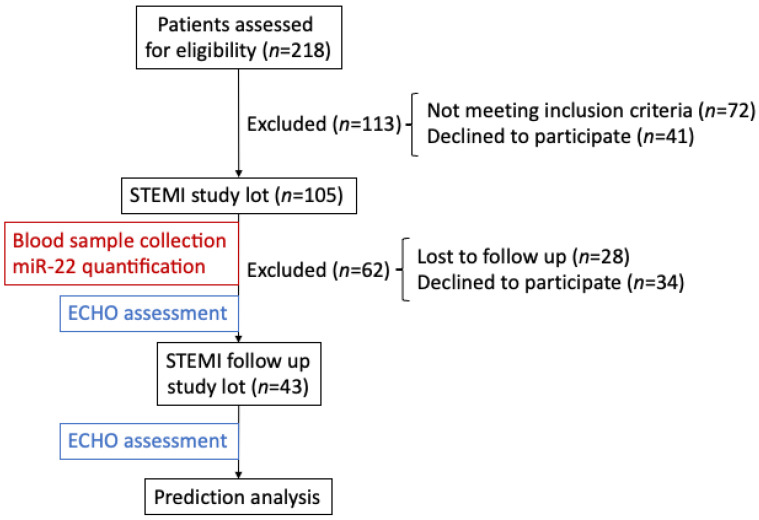
Flowchart of miR-22 analysis in the plasma of STEMI ALL and follow up cohorts.

**Figure 2 biomedicines-11-02289-f002:**
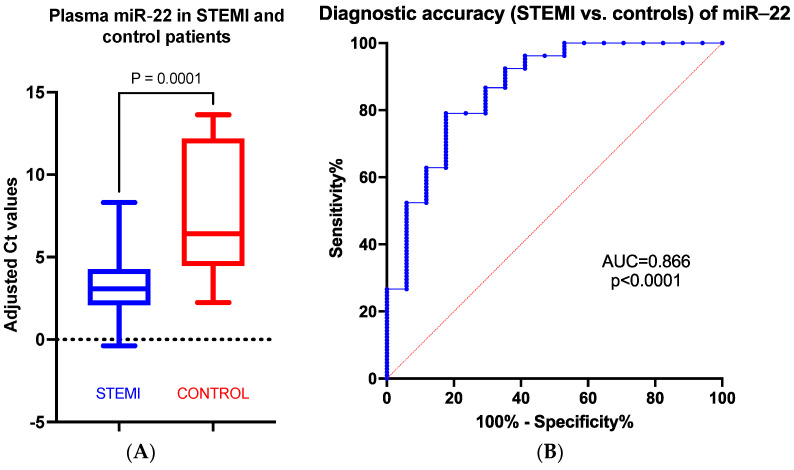
Fold change (**A**) and ROC analysis (**B**) of normalized miR-22 plasma values in STEMI patients vs. control group.

**Figure 3 biomedicines-11-02289-f003:**
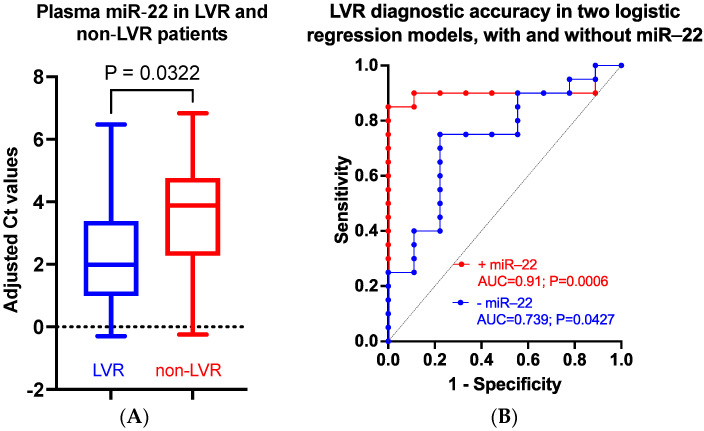
Plasma miR-22 as LVR predictor in STEMI patients. miR-22 is significantly upregulated in LVR patients vs. non-LVR patients (**A**). miR-22 significantly improves the predictive value of a multiple logistic regression model based on EFi, EDVi, CK-MB, and troponin (**B**).

**Figure 4 biomedicines-11-02289-f004:**
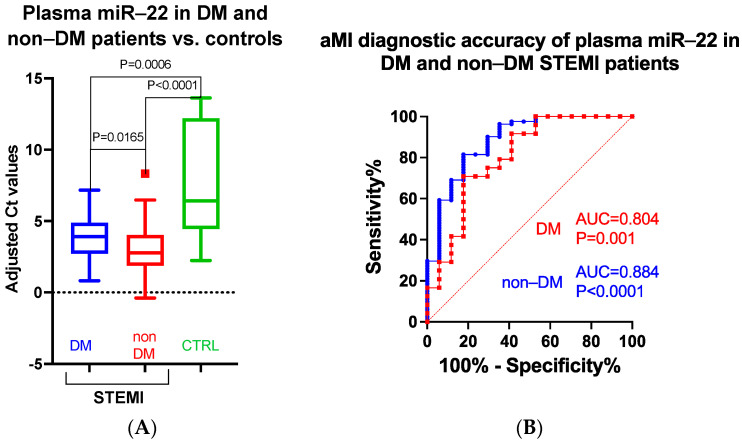
Fold change (**A**) and ROC curve (**B**) analysis of plasma miR-22 normalized (against cel-miR-39) values in DM and non-DM STEMI patients vs. controls.

**Table 1 biomedicines-11-02289-t001:** Demographic and clinical features of STEMI patients.

Variable	All (*n* = 105)	Follow Up Group (*n* = 43)	*p*
Age (years) mean ± SD	60.83 ± 12.9	57.81 ± 11.73	0.172 *
Female/Male *n*/*n*	29/76	11/32	0.253
Risk factors, *n* (%)			
Hypertension	75 (71.43)	33 (76.74)	0.509
Hypercholesterolemia	23 (11.90)	11 (25.58)	0.631
Current smoker	53 (50.48)	24 (55.81)	0.555
Obesity (BMI > 30 kg/m^2^)	31 (29.52)	16 (37.21)	0.363
Diabetes mellitus	24 (22.86)	8 (18.6)	0.569
Presentation			
CK-MB (U/L) mean ± SD	108.97± 130.3	120.31± 138.8	0.944 **
Time from symptoms onset to reperfusion (hours)	6.3 ± 3.4	6.7 ± 3.7	0.945 **
eGFR, mean ± SD	65.1 ± 23.7	72.6 ± 19.3	0.075 **
Killip Class, *n* (%)			
1	77 (73.33)	34 (79.07)	0.465
2	26 (24.76)	9 (20.93)	0.617
3	1 (0.95)	0	0.522
Burden of CAD, *n* (%)			
0	1 (0.95)	0	0.522
1	68 (64.76)	27 (62.79)	0.818
2	29 (27.62)	13 (30.23)	0.749
3	6 (5.71)	3 (6.98)	0.772
Complete revascularization *n*(%)			
0 (incomplete)	8 (7.62)	6 (13.95)	0.230
1 (complete)	80 (76.19)	32 (74.42)	0.818
2 (surgical revascularization)	7 (6.67)	1 (2.33)	0.289
3 (incomplete due to chronic occlusion or small vessels)	9 (8.57)	4 (9.30)	0.889
In-hospital death, *n* (%)	7 (6.67)	-	0.084
Type of infarction, *n* (%)			
Anterior	49 (46.67)	23 (53.49)	0.453
Inferior	51 (48.57)	19 (44.19)	0.624
Other	5 (4.76)	1 (2.33)	0.496

SD—standard deviation; BMI—body max index; CK-MB—Creatine kinase myoglobin binding; CAD—coronary artery disease; eGFR—estimated glomerular filtration rate; * unpaired heteroscedastic Student *t*-test; ** Mann–Whitney two-tailed test; all other tests are two-tailed Z tests.

**Table 2 biomedicines-11-02289-t002:** Demographic and clinical features of the follow-up LVR and non-LVR subgroups of STEMI patients.

Characteristics	LVR (*n* = 14)	Non-LVR (*n* = 29)	*p*
Age (years) mean ± SD	62.31 ± 11.32	56.6 ± 11.88	0.3545 *
Female, *n* (%)	3 (21.43)	9 (31.03)	0.509
CK-MB (U/L) mean ± SD	153.9 ± 166.6	105.4 ± 125.2	0.379 *
Killip Class, *n* (%)			
1	10 (71.43)	24 (82.76)	0.390
2	4 (28.57)	5 (17.24)	0.389
3	0	0	-
Burden of CAD, *n* (%)			
0	0	0	
1	7 (50.0)	20 (68.97)	0.226
2	5 (35.71)	8 (27.59)	0.589
3	2 (14.29)	1 (3.45)	0.190
Complete revascularization *n*(%)			
0 (incomplete)	4 (28.57)	2 (6.90)	0.055
1 (complete)	9 (64.29)	23 (79.31)	0.289
2 (surgical revascularization)	1 (7.14)	0	0.144
3 (incomplete due to chronic occlusion or small vessels)	0	4 (13.79)	0.145
Thrombolysis, *n* (%)	2 (14.29)	7 (24.14)	0.459
Cardiovascular history/risk factors, *n* (%)			
Hypertension	12 (85.71)	21 (72.41)	0.332
Diabetes mellitus	2 (14.29)	6 (20.69)	0.61
Hypercholesterolemia	2 (14.29)	9 (31.03)	0.238
Current smoker	5 (35.71)	19 (65.52)	0.066
Obesity (BMI > 30 kg/m^2^)	7 (50.0)	9 (31.03)	0.226
Anemia: Hb < 13.5 g/dL (men), <12 g/dL (women)	4 (28.57)	3 (10.34)	0.128
Medication (*n*%)			
Aspirin	14 (100)	29 (100)	-
Aldosterone receptor antagonist	11 (78.57)	25 (86.21)	0.522
Betablocker	11 (78.57)	26 (89.66)	0.328
ACEi/ARB	9 (64.29)	24 (82.76)	0.180
Clopidogrel	7 (50)	10 (34.48)	0.327
Nitrate	4 (28.57)	4 (13.79)	0.242
Statin	14 (100)	29 (100)	-
Ticagrelor	7 (50)	10 (34.48)	0.327
Type of infarction, *n* (%)			
Anterior	8 (57.14)	15 (51.72)	0.741
Inferior	6 (42.86)	13 (44.83)	0.904
Other	0	1 (3.45)	
Echo parameters			
Average change EF (%)	−10.32	12.92	0.0003 *
Average change EDV (%)	26.65	−7.28	<0.0001 *

SD—standard deviation; BMI—body max index; CK-MB—Creatine kinase myoglobin binding; CAD—coronary artery disease; eGFR—estimated glomerular filtration rate; ACEi/ARB—Angiotensin-Converting Enzyme inhibitor/Angiotensin Receptor Blocker; * Unpaired two tailed *t*-test with Welch correction; all other tests: two-tailed Z test.

**Table 3 biomedicines-11-02289-t003:** Correlation parameters between plasma miR-22 and clinical parameters in the STEMI cohort.

	Correlation Coefficient (*p*)	DM	Erythrocyte Number	Hematocrit
STEMI cohort	r	0.241	−0.228	−0.2014
	*p* value	0.0133 *	0.0193	0.0394
	95% confidence interval	-	−0.4079 to −0.0322	−0.3833 to −0.0044
Follow up cohort	r		−0.2164	−0.121
	*p* value	0.20772	0.1634	0.4397
	95% confidence interval	0.18134 *	−0.4922 to −0.0988	−0.4142 to −0.1950

*—point biserial test; all other tests are two-tailed Spearman tests.

**Table 4 biomedicines-11-02289-t004:** Correlation parameters between plasma miR-22 and echocardiographic parameters in the follow-up cohort.

	Correlation Test	EDVi	%ΔEDV	EFi	% ΔEF
LVR cohort	r	−0.3417 *	0.3784 *	−0.0669	0.1802 **
	*p* value	0.2319	0.1432	0.8204	0.5376
	95% confidence interval	−0.7384 to 0.2308	−0.1904 to 0.7570	−0.5886 to 0.4940	−0.3874 to 0.6487
Non-LVR cohort	r	0.3081 **	−0.03249 *	0.2728 *	−0.5325 *
	*p* value	0.1039	0.8671	0.1523	0.0029
	95% confidence interval	−0.0772 to 0.6133	−0.3943 to 0.3380	−0.1042 to 0.5812	−0.7522 to −0.2063

CK-MB—Creatine kinase myoglobin binding; * Pearson test; ** Spearman test.

**Table 5 biomedicines-11-02289-t005:** Multiple logistic regression analysis of LVR prediction power of EFi, EDVi, CK-MB, and troponin variables with/without miR-22 in the follow-up cohort.

Area under the ROC Curve	With miR-22	Without miR-22
Area	0.9056	0.7389
Std. Error	0.06133	0.1007
95% confidence interval	0.7854 to 1.000	0.5416 to 0.9362
*p* value	0.0006	0.0427
Negative predictive power (%)	80	66.67
Positive predictive power (%)	94.74	78.26
Tjur’s R squared	0.4592	0.1675
Hosmer-Lemeshov test (*p*)	13.04 (0.1105)	12.5 (0.1445)

**Table 6 biomedicines-11-02289-t006:** Demographic and clinical features of STEMI patients with and without diabetes.

Characteristics	Diabetic (*n* = 24)	Non-Diabetic (*n* = 81)	*p* Value
Age, years	63.63 ± 10.00	60.00 ± 13.59	0.159 *
Female, *n* (%)	10 (41.67)	19 (23.46)	0.081
Cardiovascular history/risk factors, *n* (%)			
Hypertension	21 (87.50)	54 (66.67)	0.048
Hypercholesterolemia	2 (8.33)	21 (25.93)	0.067
Current smoker	10 (41.67)	43 (53.09)	0.327
Obesity	9 (37.5)	22 (27.16)	0.327
Presentation			
CK-MB (U/L)	88.29	114.68	0.394 *
Type of infarction			
Anterior	14 (58.33)	35 (43.21)	0.194
Inferior	10 (41.67)	41 (50.62)	0.441
Other		5 (6.17)	
In-hospital death	5 (20.83)	2 (2.47)	0.0015

CK-MB—Creatine kinase myoglobin binding; * Unpaired heteroscedastic Student *t* test with Welch’s correction; all other tests are two-tailed Z tests.

## Data Availability

The data presented in this study are available in Appendix A files.

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
