# Peer review of "Plasma hsa-miR-22-3p Might Serve as an Early Predictor of Ventricular Function Recovery after ST-Elevation Acute Myocardial Infarction"

_biomedicines, 2023, doi:10.3390/biomedicines11082289_

Round 1

Reviewer 1 Report

Mir-RNAs are important  molecules and  potential biomarkers and  targets for  treatment. A nice  study analyze  mir-22 serum  level in patients with STEMI  and  LV remodeling.  Study is  well done. The  main limitation as mentioned in limitation section,  were small numbers, only half  of the  patients could  be  followed up. The paper requires several corrections  before publication.

Namely:

1.       The  title and  methodology. Did you analyze mir-22-3 p or  22-5p?

2.       Several patients received thrombolysis before PCI.  Have  to be  mentioned. Could  it influence the  results?

3.       Blood samples were derived first day of presentation, before or  after  PCI?

4.       What was the  method of  hemolysis exclusion in samples

5.       Need more data about  control group

6.       Table  1   could  include  data about  eGFR and  previous  atherosclerotic complications (stroke, PAD  etc – if  any)

7.       Table 2  should  include time  to reperfusion and  CAD  burden (1 vs  3  vessel disease), Killip etc.

8.       English requires proofreading (eg. blood hypertension, line  212)

English requires proofreading

Author Response

Dear Reviewer 1,

Thank you very much for your letter and for taking the time and effort to evaluate our manuscript. We appreciate all your comments and suggestions, which helped us in improving the quality of the manuscript; the revised portions are marked in yellow.

The main corrections in the paper and the point-by-point responses to your comments  are outlined (in italics) as follows:

Mir-RNAs are important  molecules and  potential biomarkers and  targets for  treatment. A nice  study analyze  mir-22 serum  level in patients with STEMI  and  LV remodeling.  Study is  well done. The  main limitation as mentioned in limitation section,  were small numbers, only half  of the  patients could  be  followed up.

Thank you for your kind comments on our work.

The paper requires several corrections  before publication.

Namely:

  1. The title and  methodology. Did you analyze mir-22-3 p or  22-5p?

We apologize for this oversight. We have modified accordingly the title and operated the due changes in the manuscript: the title (line 2), the abstract (line 20), and the introduction (line 74).

  1. Several patients received thrombolysis before PCI. Have  to be  mentioned. Could  it influence the  results?

The requested data and the comparative analysis have been included in Table 2: there are no significant differences between LVR and non-LVR subgroups. These data do not influence the results.

  1. Blood samples were derived first day of presentation, before or after  PCI?

All blood samples were retrieved on admission day, before PCI. A statement regarding this aspect was added to the Materials and Methods section, lines 116-117.

  1. What was the method of  hemolysis exclusion in samples

We assessed the presence of hemolysis spectrophotometrically by measuring absorbance at 414nm. A brief statement (lines 117-118) has been included in the Materials and Methods section.

  1. Need more data about control group

All the data we have on the control group patients have been included in Supplementary File 1, Sheet “CONTROLS”.

  1. Table 1   could  include  data about  eGFR and  previous  atherosclerotic complications (stroke, PAD  etc – if  any)

As requested, we have added an eGFR column in Supplementary File 1. There are no significant correlations with miR-22 plasma levels.

  1. Table 2 should  include time  to reperfusion and  CAD  burden (1 vs  3  vessel disease), Killip etc.

The data on the time from symptoms onset to reperfusion have been already included in Supplementary File 1 under the column “DEBUT/ ONSET OF SYMPTOMS (hours)”; for clarity, we have renamed it “TIME FROM SYMPTOMS ONSET TO REPERFUSION”. Furthermore, we have included it as such also in Supplementary File 2 (the follow-up group).

As requested, we included in both Supplementary Files 1 and 2 the Killip Class, CAD burden data, and reperfusion data. None of these data correlate with miR-22 plasma levels.

  1. English requires proofreading (eg. blood hypertension, line 212)

We have perused the manuscript and, hopefully, have identified all typos and grammar mistakes.

Reviewer 2 Report

This interesting work is about the relationship between ventricular recovery and microRNA 22. This data suggests that miR-22 might be used as a predictor of ventricular function recovery in STEMI patients.

1. The work enrolled 105 patients, but follow-up echo was only available in 43 patients. The title discusses the issue of ventricular recovery from limited data. Therefore, the data of the other 62 STEMI patients may not enroll in this study.

2. How about the interval of door-to-ballon time and complete revascularization for the level of microRNA?

3. Due to limited patients, the author may consider to separate the analysis and discussion about DM population.

Author Response

Dear Reviewer 2,

Thank you very much for your letter and for taking the time and effort to evaluate our manuscript. We appreciate all your comments and suggestions, which helped us in improving the quality of the manuscript; the revised portions are marked in yellow.

The main corrections in the paper and the point-by-point responses to your comments are outlined (in italics) as follows:

This interesting work is about the relationship between ventricular recovery and microRNA 22. This data suggests that miR-22 might be used as a predictor of ventricular function recovery in STEMI patients.

Thank you for your nice comments on our work.

  1. The work enrolled 105 patients, but follow-up echo was only available in 43 patients. The title discusses the issue of ventricular recovery from limited data. Therefore, the data of the other 62 STEMI patients may not enroll in this study.

The reason why we decided to keep the data on all patients initially enrolled, is related to the need to prove that the follow-up sub-group is representative of the entire STEMI group of patients and that there are no significant changes in the clinical and paraclinical parameters between the STEMI ALL and follow-up that might have an impact miR-22 plasma levels.

  1. How about the interval of door-to-ballon time and complete revascularization for the level of microRNA?

Unfortunately, the hospital's electronic records do not contain data on door-to-balloon time and complete revascularization. However, since we are well aware of the importance of this kind of parameter, we decided to use as a proxy the time from symptoms onset to reperfusion, which we show is not correlated with plasma miR-22 levels.

  1. Due to limited patients, the author may consider to separate the analysis and discussion about DM population.

This is a very interesting observation, and we have extensively discussed this issue with our clinician colleagues.

At this point, we feel that, given that DM is known to influence LVR and LV dynamics (and we found it strongly impacts plasma miR-22 levels), one should properly analyze its impact on miR-22 ability to diagnose aMI and predict LVR. Both the results and discussion sections have dedicated, separate paragraphs about the DM influence on LVR and miR-22's ability to predict LVR.

Round 2

Reviewer 1 Report

The  manuscript is improved substentially.

Several mild  issues that  were  addresed in previouse review could be  corrected. Namely,

1.  I reccomend to clarify the  control group in the  text  body (e.g. outpatient  or  inpatient  patients etc)

2. GFR  data  could  be  included into table  1   in the  manuscript

Author Response

Dear Reviewer,

Thank you once again for taking the time and effort to evaluate our manuscript; your comments and suggestions helped us significantly improve the quality of the manuscript.

The  manuscript is improved substentially.

Several mild  issues that  were  addresed in previouse review could be  corrected. Namely,

1. I reccomend to clarify the  control group in the  text  body (e.g. outpatient  or  inpatient  patients etc)

All individuals in the control group are outpatients; the matter has been clarified accordingly in the Materials and Methods section (line 103).

2. GFR  data  could  be  included into table  1   in the  manuscript.

As requested, GFR data were included in Table 1.

Reviewer 2 Report

The authors clearly respond to the comments. I did not have other comments.

Author Response

We thank the reviewer for her/his work and patience in revising our paper; your comments helped us improve our manuscript.